# Convergence of Artificial Intelligence and Neuroscience towards the Diagnosis of Neurological Disorders—A Scoping Review

**DOI:** 10.3390/s23063062

**Published:** 2023-03-13

**Authors:** Chellammal Surianarayanan, John Jeyasekaran Lawrence, Pethuru Raj Chelliah, Edmond Prakash, Chaminda Hewage

**Affiliations:** 1Centre of Distance and Online Education, Bharathidasan University, Tiruchirappalli 620024, India; 2Cardiff School of Technologies, Cardiff Metropolitan University, Cardiff CF5 2YB, UK; 3Edge AI Division, Reliance Jio Platforms Ltd., Bangalore 560103, India; 4Research Center for Creative Arts, University for the Creative Arts (UCA), Farnham GU9 7DS, UK

**Keywords:** artificial intelligence, neuroscience, deep neural network, neurological disorders

## Abstract

Artificial intelligence (AI) is a field of computer science that deals with the simulation of human intelligence using machines so that such machines gain problem-solving and decision-making capabilities similar to that of the human brain. Neuroscience is the scientific study of the struczture and cognitive functions of the brain. Neuroscience and AI are mutually interrelated. These two fields help each other in their advancements. The theory of neuroscience has brought many distinct improvisations into the AI field. The biological neural network has led to the realization of complex deep neural network architectures that are used to develop versatile applications, such as text processing, speech recognition, object detection, etc. Additionally, neuroscience helps to validate the existing AI-based models. Reinforcement learning in humans and animals has inspired computer scientists to develop algorithms for reinforcement learning in artificial systems, which enables those systems to learn complex strategies without explicit instruction. Such learning helps in building complex applications, like robot-based surgery, autonomous vehicles, gaming applications, etc. In turn, with its ability to intelligently analyze complex data and extract hidden patterns, AI fits as a perfect choice for analyzing neuroscience data that are very complex. Large-scale AI-based simulations help neuroscientists test their hypotheses. Through an interface with the brain, an AI-based system can extract the brain signals and commands that are generated according to the signals. These commands are fed into devices, such as a robotic arm, which helps in the movement of paralyzed muscles or other human parts. AI has several use cases in analyzing neuroimaging data and reducing the workload of radiologists. The study of neuroscience helps in the early detection and diagnosis of neurological disorders. In the same way, AI can effectively be applied to the prediction and detection of neurological disorders. Thus, in this paper, a scoping review has been carried out on the mutual relationship between AI and neuroscience, emphasizing the convergence between AI and neuroscience in order to detect and predict various neurological disorders.

## 1. Introduction

Artificial intelligence (AI) is a field of computer science that deals with the simulation of human intelligence in machines [1] such that the machines have problem-solving [2] and decision-making capabilities similar to that of a human brain [1]. AI-based systems are trained with huge amounts of data so that they learn how to perform a task. Later, the systems use the learned knowledge to analyze the unknown inputs to produce the desired outcome. The unique potential of this field is that it can analyze massive amounts of data quickly without human intervention. For example, during the time of the COVID-19 pandemic, an unprecedented number of individuals were infected, and AI-based systems were programmed to automatically detect the presence of COVID-19 in individuals [3,4]. Additionally, the performance of AI-assisted CT scan imagery analysis is equivalent to an expert radiologist [5]. A revolution in hardware technologies saw a shift from traditional machine learning to deep learning in AI, and made various popular applications, like expert systems, computer vision, natural language processing (NLP), speech recognition, and image classification, come into existence [6]. Still, hardware technologies advance towards neuromorphic hardware so that the power consumed by AI systems would be lower, keeping in mind the energy efficiency of the human brain [7]. In short, AI enables machines to solve complex problems and make decisions, both intelligently and intuitively.

Neuroscience is the scientific study of the structure and cognitive functions of the brain regarding processing data, making decisions, and interacting with the environment [8]. It combines different disciplines, such as physiology, anatomy, molecular biology, cytology, psychology, physics, computer science, chemistry, medicine, statistics, mathematical modeling, etc. [9]. Neuroscientists not only focus on the study of the brain for cognitive functioning but also investigate the whole nervous system to get a comprehensive understanding of different neurological, psychiatric, and neurodevelopmental disorders [10]. Neuroscience reveals which parts of the human nervous systems are likely to get affected by diseases, disorders, and injuries, and thereby helps with effective treatments. Another key point to be mentioned here is that the advancement in neuroimaging technologies has largely contributed to the understanding of the structure and function of the brain [11,12]. Actually, the evolution of neuroscience has been driven by advancements in tools and technologies, which have enabled the study of the brain both at high resolution by examining genes, molecules, synapses, and neurons and at low resolution with whole-brain imaging [13]. Due to having many hidden layers, convolutional neural networks have found applications in radiology regarding analyzing images with high-level reasoning for detection and prediction tasks [14]. In addition, the retrieval of meaningful insights and their storage, manipulation, visualization, and management are facilitated by computer-based neuroimaging tools [15]. 

The quote, “At some point in the future, cognitive neuroscience will be able to describe the algorithms that drive structural neural elements into the physiological activity that results in perception, cognition, and perhaps even consciousness”, from Gazzaniga, one of the founders of cognitive neuroscience, which is discussed in [16], the author explains that AI can benefit from the study of neural mechanisms of cognition, as the structure of the cortex suggests that the implementation of cognitive processes in the brain occurs with networks and systems of networks based on the uniform local structures of layers, columns, and basic local circuits as building blocks. Additionally, the author describes that the collaboration between AI and neuroscience can produce an understanding of the mechanisms in the brain that generate human cognition because AI and computer power can produce large-scale simulations of neural processes that generate intelligence. From the above statement, it is clear that neuroscience lays the foundation for the design of artificial neural networks (ANNs), which consist of nodes structured in input, hidden, and output layers. Hence, it is realized that there is a complementary relationship between neuroscience and AI, irrespective of their different end purposes or goals. The complementary relationship between AI and neuroscience has gained momentum as they advance by helping each other and ultimately leading to the development of useful applications for the detection and diagnosis of various neurological disorders.

In the past few years, some reviews have investigated the mutual relationship between AI and neuroscience. In [17], the authors performed a review with a major focus on the role of neuroscience in advancing AI research, along with highlighting the applications of AI for the advancement of neuroscience. The authors discussed the inspiration from neuroscience for the development of new types of algorithms and the usefulness of neuroscience for the validation of AI techniques. In [18], the authors reviewed the relationship between neuroscience and AI and the recent advancements in four areas, namely, AI models of working memory, AI visual processing, AI analysis of big neuroscience datasets, and computational psychiatry. In [19], the authors conducted a review on how neuroscience served as the source of guidance in constructing ANNs to deep networks and their further transformations. In [20], the authors reviewed how reinforcement learning correlates with neuroscience and psychology. In [21], the authors discussed the sharing relationship between AI and neuroscience by highlighting the intersections of biological vision and AI vision networks. The research work [22] also discusses how AI and neuroscience obtain benefits from each other when dealing with massive knowledge bases.

In short, neuroscience and AI are mutually intertwined. AI has a wide range of applications in various domains with the common aspect of making machines intelligent like humans so that various complex tasks, such as speech recognition, game applications, selfdriven car applications, intelligent traffic management, robotics-based surgery, image and video analytics, NLP, etc., can be performed effectively. By studying the structure and functioning of the brain, neuroscience helps in the effective detection and diagnosis of various neurological disorders. 

With the above background in mind, in this work, a scoping review has been performed on the mutual relationship between neuroscience and AI by describing how one field helps the advancement of the other. With its ability to analyze complex and huge amounts of data and extract patterns, AI is extensively used for the early detection and prediction of neurological diseases. Neuroscience uses knowledge to predict and detect various neurological diseases. So, with the above perspective, in this paper, the convergence of AI and neuroscience is realized in the detection of neurological disorders. Thus, a special focus has been given to the applications of AI for the detection and diagnosis of various neurological disorders. In addition, a brief overview of the contributions of AI and AI-based tools to the effective analysis of neuroimages is presented.

The rest of the paper is organized as follows. Section 2 presents the objective. Section 3 describes the review method. Section 4 describes how neuroscience has helped in the design of AI. Section 5 presents how AI helps the advancement of neuroscience, along with a short description of AI in neuroimaging analysis. Section 6 describes, in detail, the applications of AI in the prediction and diagnosis of various neurological disorders in a categorical manner. Section 7 describes the various challenges and future directions of research. Section 8 concludes the work.

## 2. Objective

With respect to neuroscience, AI helps to simulate the brain so that neuroscientists can test their hypotheses. Additionally, explainable AI methods help with the interpretation of large multimodal datasets and enable neurologists to detect neurological disorders early [23]. For example, deep neural networks and deep reinforcement learning allow neuroscientists to know how impulses from the brain are communicated to other parts of the body, which helps in the early detection of movement-related disorders like paralysis. As mentioned earlier, the convergence of AI and neuroscience occurs in order to detect and predict different neurological disorders. From this perspective, the objective of this review has been set to investigate the literature that describes the complementary relationship between AI and neuroscience and to retrieve how these two fields converge to enable useful applications in the prediction and detection of different neurological disorders. 

The research questions of the current study include:How significant is the relationship between AI and neuroscience?How do other existing surveys focus on this topic?How does neuroscience inspire the design of AI?How does AI help in the advancement of neuroscience?What are the applications of AI in neuroimaging methods and tools?How does AI help in the diagnosis of neurological disorders?What are the challenges associated with the implementation of AI-based applications for neurological diseases?What are the directions for future research?

## 3. Review Method

The publications relevant to the objective of the paper have been collected from different data sources, namely, Frontiers in computer science, Web of Science, PubMed, Scopus, arXiv, Springer, and IEEE, as well as from Google using keyword querying methods. The search has been performed in iterations. Different keywords, namely “artificial intelligence and neuroscience”, “relationship between AI and neuroscience”, “applications of neuroscience for AI”, and “applications of AI for neuroscience”, were used in the initial iterations, and in each iteration, the titles of the retrieved articles were manually analyzed to determine further optimized keywords, such as “natural and artificial intelligence”, “inspirations of neuroscience for AI”, “interplay between AI and neuroscience”, “sharing relationship between AI and neuroscience”, “neuroimaging in the era of AI”, etc., for subsequent iterations. The above iterative querying resulted in an initial set of 260 articles (235 from scientific databases and 25 hyperlink records from Google) identified for the study. From the initial set of articles, duplicate records (17 scientific records) were removed. The abstracts of the scientific articles were analyzed, and those abstracts that were not related to the proposed objective (another 12 records) were removed during screening. The remaining full articles (206 scientific articles and 25 hyperlink records) were assessed, and the articles that did not contain useful information for the current study and that were not directly related to the current scope were eliminated. The preferred reporting items for systematic reviews and meta-analyses (PRISMA) flow diagram of the review method is shown in Figure 1. A collection of 185 representative publications (173 records from scientific databases and 12 hyperlink references from Google) were considered for the current study. It has been categorized according to the research questions. The categorization is given in Table 1 and Figure 2. The publications were carefully analyzed, and the findings are presented in the subsequent sections.

## 4. Neuroscience for AI

Conventional AI systems are based on manually extracted features, and they have a limited capability of processing the data in their original form [24]. They are based on hard-coded statements in formal languages that can be reasoned out by a computer using logical inference rules, and these systems have a limited capability of solving complex tasks, such as object recognition, text analytics, and image classification [25,26]. In contrast to the conventional AI systems, the design of ANN was inspired by the biological network of neurons in the human brain, leading to a learning system that is far more capable than the conventional learning models [27]. 

Deep learning allows machines to be fed with raw data and enables the machines to undergo automatic discovery regarding the features required for various tasks, such as object detection and speech recognition. When one looks into the symbiotic relationship between AI and neuroscience, it is clear that the theory of neuroscience helps AI in two ways; first, in the design and development of different variants of ANN and its learning methods, and secondly, in the validation of existing AI-based models [28]. If a known algorithm is discovered to be implemented in the brain, it would be a strong support for its viability as a component in large general intelligence systems [29]. The inspiration from neuroscience for advancements in the design of AI is described in the subsequent subsections.

### 4.1. ANN

ANN is inspired by neurons in the human brain. An ANN is composed of several interconnected units, or artificial neurons, which work in parallel [30]. Each artificial neuron is linked to several others and can transmit signals along these connections. The strength of these connections will be modified during learning, and the learned knowledge will be stored in weights. The weights of connections are updated similarly to the mechanism of Hebbian learning, as described in [31]. Further, the invention of the microscope helped in understanding the structure of the brain and the connections among neurons. The structure of the brain and concept of Hebbian learning enabled AI scientists to design a simple ANN called perceptron. Frank Rosenblatt, who designed the perceptron in the 1950s [32,33], developed a simple rule for classifying the output, as shown in Figure 3.

The perceptron takes one or more binary inputs and produces a desired binary output. The mapping of inputs to the corresponding outputs is basically decided by the application in hand. Let $x_{1},x_{2},x_{3},\ldots x_{i}$ denote the binary inputs and $w_{1},w_{2},w_{3},\ldots w_{i}$ denote their weights, respectively. In the hidden layer, the rules for transforming the inputs into a single binary output are defined. At first, each input is multiplied by its weight, and then all multiplied terms are added. Then a bias (constant) is added. The rules for determining the output are defined through the activation function, such as the binary step function, defined in Equations (1) and (2):(1)$$f(x)\;=\;0,\;if\;\sum_{i}w_{i}\;\times \;x_{i}\;+\;bias\;\leq \;0$$
(2)$$f(x)\;=\;1,\;if\;\sum_{i}w_{i}\;\times \;x_{i}\;+\;bias\;>\;0$$

The binary output is denoted by y, which is given by f(x). This mapping of input to output can be compared with the human nervous system, in which, at any instant, if a neuron has some substantial threshold of any neurotransmitter, it sends the neurotransmitter to the receptors of the subsequent neuron via synapses. According to the neurotransmitter sent, the second neuron performs its activity. Here, similarly to the human brain, which learns by varying the connection strengths between neurons, which involves removing or summing connections between neurons [34], in ANN, the weights of connections are modified during learning. The weights and biases are adjusted to produce the desired output for the given inputs. When the inputs and outputs are binary, at times, there will be a sudden flip in the output states from 0 to 1, even for small changes in the weights and bias. In order to avoid this situation, sigmoid neurons are defined where the inputs are not binary. An input is assigned with a continuous value from 0 to 1. The Sigmoid function is computed using the formula given in Equation (3).
(3)$$y\;=\;\sigma (\sum_{i}w_{i}\;\times \;x_{i}\;+\;bias)$$

In Equation (3), σ, the sigmoid function, helps to smooth the sudden step change that may occur in the output of the binary perceptron. The inputs and weights are real values. If the predicted output of perceptron is found to be the same as the expected output, the functioning of the perceptron is considered satisfactory; otherwise, it must be trained, and the weights have to be adjusted to produce the desired output. Once the training is complete, the perceptron is able to produce an output of 0 for input values from 0 to 0.5 and 1 for input values from 0.5 to 1.

### 4.2. Multilayer Perceptron (MLP)

The single-layer perceptron model has limitations in solving problems that involve the XOR function. Therefore, perceptron networks have been modified with more than one hidden layer with many neurons stacked together, called MLP. In MLP, the outputs of one layer feed the next layer. This continues through the hidden layer and goes up to the last layer, in which the arrived-at values are altered by the appropriate activation function to produce the expected results. MLP is trained using the backpropagation learning method, which consists of two passes through the MLP. In the forward pass, the output of the network is computed for the given inputs and initial weights. In the backward pass, the difference between the actual output and the expected output is computed as the error, and the error is propagated from the output to the inputs with the intention of updating the weights using gradient descent [35]. Here, the machine learning experts intentionally introduced the “backpropagation of error” to modify the weights so that the error would be reduced. As the backpropagation method does not need any set parameters other than the inputs, the method is simple. The method uses a gradient descent approach with a chain rule to update the weights in an iterative manner. One key point to be noticed here is that the backpropagation of error in AI has its equivalence in biological neural networks, as discussed in [36] where the authors discussed the fact that cortical networks (with simple local Hebbian synaptic plasticity) implement efficient learning. The same idea was supported in [37]. Further, in MLP, the design of the input and output layers is simple and straightforward. But the design of the hidden layer is based on domain expertise and application requirements. 

### 4.3. Recurrent Neural Network (RNN)

As discussed in [18], neuroscience provides inspiring methods for constructing ANNs with working memories, which is mediated by the persistent activity of neurons in the prefrontal cortex [38] and other areas of the neocortex and hippocampus [39], called recurrent neural networks (RNNs). 

The feature of working memory in the human brain inspired the design of RNNs, which store the recent past output in an internal memory structure. The architectures prior to RNNs (like feed-forward networks) can produce outputs that correspond only to the current inputs. In RNNs, the past output is fed back to the input, which helps in the prediction of the next output, as shown in Figure 4.

Working memory is a key cognitive capacity of biological agents [40], and this capacity is required for processing sequential data [41]. Additionally, conventional RNNs can retain only the recent past output, whereas a special kind of RNN called a long, short-term memory (LSTM) network can handle long-term dependencies and is useful in video classification, speech recognition, and text summarization. The architecture of an LSTM network consists of memory cells having connections and gates. The gates are of three kinds: input, output, and forget gates. The flow of input is controlled by the input gates; the cell activation of the remaining network is controlled by output gates, and the forget gates decide how much previous data needs to be forgotten and how much has to be remembered. The design of an RNN and its variants was inspired by the circuit architecture of the midbrain attention network [42]. 

### 4.4. Convolutional Neural Network

The availability of high-performance computing devices, graphics processing units (GPU), and software technologies ushered in the second generation of deep neural networks, called convolutional neural networks (CNNs). The connectivity pattern between neurons in CNNs was inspired by the architecture of the brain’s ventral visual stream [43]. The ventral visual stream has two features: retinotopic and hierarchical. Here, retinotopic refers to the organization of visual pathways according to the way the eye takes visual information in, and hierarchical refers to how specific areas of the cortex perform increasingly complex tasks in a hierarchical manner. For example, as far as an object is concerned, the cortex identifies only the outlines, edges, and lines of the object (which is of less complexity) and then identifies the big parts of the objects. The visual information that enters one’s eyes travels through the neurons of the brain and is perceived as the concerned objects after passing through different stages of increasing complexity, such as from edges, lines, curves, etc., to, say, for example, faces or full bodies. The construction CNNs was completely inspired by the above logic of transforming the visual input from simple features into increasingly complex features and then to object recognition [44].

The typical architecture of a CNN usually includes three layers of increasing complexity: a convolutional layer, a pooling layer, and a fully connected layer. Each layer has different parameters that must be optimized for performing a specific task. The convolutional layers are the layers in which filters are applied to the original image. Typical filters, which are composed of small kernels of size 3 × 3 or 5 × 5, move across the image from top left to bottom right and perform the assigned mathematical operation. For example, a neuron in the first layer performs an elementwise multiplication between the weights and pixel values and then returns the total sum of the result. In a convolutional layer, the number of kernels and the kernel size are given as the input parameters. This layer extracts features. The pooling layer performs aggregation functions, such as picking up the maximum value or average value from the result of the previous layer and, thus, reduces the dimension involved in the network. It aggregates the information contained in the features. The final, fully connected layer is used to smooth the obtained information and ultimately performs the task of classification or detection.

### 4.5. Reinforcement Learning (RL)

An important topic of research in AI is reinforcement learning, which refers to learning through rewards and punishments. In reinforcement learning, an agent interacts with the environment and learns to act within it. Here, the agent itself learns to repeat certain tasks based on rewards and avoids certain tasks based on penalties. It learns automatically from the feedback without any guidance in a trial-and-error basis model. It develops mathematical models to increase rewards. This basic concept of reinforcement learning is being applied to applications such as aircraft control, robot motion control, computer gaming, and robotics for industrial automation. 

The reinforcement learning concept is inspired by the basic mechanism of the brain. For example, dogs were trained to have food after hearing a bell. The dogs expected food every time the bell rang and started to salivate at the sound of the bell. This led the dogs to salivate at the mere sound of the bell, even without food [referred to as Pavlovian conditioning]. This happened due to the virtue of the relationship between the bell and the food. The key point is that the organisms start learning by the error (called prediction error) between the expected outcome (i.e., getting food at the ring of bell) and the actual outcome (i.e., whether the dog really received food at the ring of the bell). As far as learning is concerned, organisms learn by experience on a trial-and-error basis to minimize error. The animal learns by understanding the error between its predicted outcome and the actual outcome. Here, the action of the dog is based on its knowledge learned from experience, and this is referred to as instrumental conditioning or trial-and-error learning [45,46]. In biological terms, the dopaminergic neurons that are located at the site of the striatum encode the prediction error in terms of temporal differences and continue to learn to choose the best actions toward the maximum reward [47]. Here the key point is that the ideas in reinforcement learning of AI are inspired by animal learning, psychology, and neuroscience [48]. 

In reinforcement learning in AI (shown in Figure 5), an intelligent agent interacts with the environment that is in some specific state. There are various actions available to the agent. The agent performs some action, and the action may change the state of the environment to some other state. Here, the agent is given a reward if their action is correct and is given a penalty if there is a mistake. Additionally, the agent consists of a policy and reinforcement algorithm. Policy represents a mapping that selects actions based on the given state of the environment. Here, the policy is like a key-value pair in a look-up table, in which, against each state, the action to be taken is stored. More important is that the learning algorithm updates the policy according to the states, actions, and rewards. The purpose of learning is to maximize the rewards. 

For example, in a computer-based chess game, if a chess piece is chosen for a particular chess grid and if the position is likely to be able to attack the opponent’s pieces, then this position is good (i.e., reward); otherwise, the position is weak if it can be attacked by the opponent (i.e., the gamer will lose the piece). Here, the intelligent agent (computer) must understand the state of the environment and the possible actions. In addition to the simple reinforcement model discussed above, the intense feeling generation of dopamine for rewards inspired distributed reinforcement learning, where the agent tends to find multiple factors that cause rewards and punishments in a wide spectrum of optimistic and pessimistic ways [49].

### 4.6. Deep Reinforcement Learning

In the case of reinforcement learning in AI, if the environment is simple, then a look-up table, like a simple policy, may be sufficient for implementation. But if the environment is complex (as in video games), a simple look-up table may not be sufficient, and the policy would be a parameterized function that can be derived by deep neural networks (as in Figure 6), as discussed in [50]. The deep neural network provides gradient-descent-based nonlinear mapping between the current states of the environment and the available actions. In modern deep reinforcement learning, learning that is based on complementary learning system (CLS) theory [51] is being used. As mentioned in [52], the human brain performs two complementary tasks: one is generalization (from the experiences performed by the neocortex), and the other is learning from specific events happening at specific times and locations via the hippocampus. Additionally, the knowledge gained by the hippocampus from rapid learning is stored in episodic memory and integrated into the consolidated information in the neocortical system for long-term storage [53]. On par with instance-based learning in biological neural networks, in deep reinforcement learning in AI, the generalization ability of neural networks is integrated into the best past outcomes of individual instances that are stored in episodic memory [54]. The concept of reinforcement learning, in which generalized learning is integrated into learning from instances, is shown in Figure 6 [51].

### 4.7. Spiking Neural Network (SNN)

Though existing deep neural networks have led to the machine recognition of images, sounds, and text, they consume huge computing resources, which may not be available in a resource-constrained environment, such as with edge devices [55]. So, as an alternative, spiking neural networks (SNNs) are being evolved. In biological neural networks, a neuron passes its electric signal to another only when its membrane potential hits a particular threshold. The same concept is introduced in SNN models, where a neuron only fires if there is a threshold breakthrough [56]. In contrast to traditional neural networks in which the signal transition takes place continuously as part of the propagation cycle, a spiking neuron fires exactly at the point of hitting a threshold, as shown in Figure 7. Spiking neurons are more energy efficient and can provide the means for implementing energy-efficient tools for modeling complex information processing [57].

Thus, neuroscience inspires the design of different ANN architectures and learning paradigms. In addition, when AI-based systems are equipped with psychological insights, such as empathy, trustworthiness, etc., the system will provide enhanced human-machine interaction. 

## 5. AI for the Development of Neuroscience

The primary strength of AI is its potential to analyze huge amounts of complex data and extract hidden patterns from within it. The signals from the brain are complex, and AI serves as the most appropriate choice for extracting inferences and patterns. As mentioned in [58], high-performing AI systems have been applied to the models that drive hypotheses about brain function. Basically, AI helps with analyzing cognition processes by producing large-scale simulations of the neural processes that generate intelligence. For example, as described in [59], an IBM research group represented 8 × 10^6^ neurons and 6400 synapses per neuron in the IBM Blue Gene processor, which can serve as a research tool for the study of neuroscience. With this research tool, neuroscientists can test their hypotheses and analyze the results from simulations before investing huge resources in actual testing with animals.

### 5.1. AI helps in Brain Computer/Machine Interface (BCI)

BCI represents the direct communication between the brain and an external device, such as the computer of a robotic arm. The device acquires the brain signals, analyzes them, and converts them into commands that carry out the intended actions. The aim of a BCI is to provide useful functions to people who have been affected by neuromuscular disorders, such as amyotrophic lateral sclerosis, cerebral palsy, stroke, or spinal cord injuries [60]. A simplified block diagram of a BCI is shown in Figure 8. Usually, the signals from the brain are acquired using semi-invasive techniques, such as electrocorticography, or with noninvasive techniques, such as *MagnetoEncephaloGraphy (MEG)*, positron emission tomography, *ElectroEncephaloGraphy (EEG)*, or any other semi-invasive or noninvasive method. The acquired data are processed for noise removal and amplification. The amplified signal is converted into a digital format. The digital data are preprocessed. The features are extracted from the preprocessed data and are then classified. Ultimately, commands are generated according to the classification. These commands are given to devices, such as a robotic arm, which helps a person with paralysis make movements. 

As discussed in [61], when using AI classifiers, a two-dimensional cursor, along with clicking abilities, helps a patient to type and use a computer like a healthy person. Further, as discussed by the authors, an implant called BrainGate uses a cursor control to control limb movements. Additionally, the authors discussed the use of AI for the effective control of prostheses, which are external devices that replace a missing part of the body. The review article [62] describes a wide range of AI-assisted BCI applications.

### 5.2. AI helps in Stimulation Studies and in the Analysis of Neurons at the Genetic Level

Measuring the gene expression of a specific cell type is crucial for determining the cellular phenotype that triggers a specific neurodegenerative disease. Here, AI helps in creating simulated models and facilitating neuron analyses from a genomic perspective. This helps in obtaining in-depth information about impulse formation in the brain and its transfer throughout the body. Such deeper studies help in identifying a specific cellular phenotype that causes a particular disease [63]. Case studies, such as the Blue Brain Project [64], aim to provide AI-based stimulated models that serve as virtual research tools to understand the characteristics of the human brain and their relationships to large-scale cognitive functioning [65].

### 5.3. AI helps in the Study of the Connectome

A connectome is the complete map of the neural connections in a brain [66]. Functional magnetic resonance imaging (fMRI) and diffusion MRI (DMRI) are used for the in-vivo analysis of the functional and structural connectomes of the human brain. When compared with standard medical images, which are of a grid-like structure, the connectome data look like network-type data, where each brain region is represented as a node in the network that may be connected either structurally and/or functionally to any other region (i.e., the other node), with the connections represented by weighted edges [67]. In this work, the authors described the need for specific training of machine learning algorithms to deal with network-structured data and its applicability for the early diagnosis of autism, delayed motor and cognitive development in preterm infants, and other neurodegenerative diseases, namely Parkinson’s disease (PD) and Alzheimer’s disease (AD). Additionally, the use of AI together with connectome data for the differential diagnosis of, say, for example, early or late mild cognitive impairment (which is a challenge to traditional image analysis techniques) has been described. Furthermore, the authors discussed the potential of AI to reveal important connections and subnetworks, facilitating an understanding of disease etiology.

### 5.4. AI helps in Neuroimaging Analysis

ANN architectures resemble the structure of the cognitive functions of humans, such that these architectures can learn the composition of features, which ultimately results in recognition of a hierarchy of structures in the data [68]. The brain uses high-dimensional sensory datasets of enormous complexity, which are increasingly difficult to analyze; only with machine learning techniques can the reconstruction of such datasets become real [69]. Additionally, as mentioned in [70], radiological imaging data grow at an unbalanced rate with respect to the number of radiologists available. In order to meet the workload, the available radiologists have to work with increased productivity, and a radiologist is likely to interpret an image once every 3–4 s in an 8 h workday [71], with errors being inevitable in constrained situations [72]. Deep neural networks can automatically extract features from complex data [73], and they produce results with an accuracy equivalent to that of a radiologist while performing segmentation [74] and detection [75]. In addition to downstream tasks, such as the detection, segmentation, characterization, and diagnosis of disorders, deep neural networks are also used to perform different upstream tasks, like removing noise, improving the resolution, normalizing images, lowering radiation and contrast dose, speeding up image acquisition, image reconstruction, image registration, etc. [76,77]. Some of the common AI use cases in the neuroimaging analysis are given below.

Speeding up MRI data acquisition—compressed sensing technology is being used as a standard technique to reduce the time involved in the data acquisition of MRI [78] and AI helps to reduce aliasing and improve the resolution of compressed images in MRI [79];Improving signal-to-noise ratio—MRI images often suffer from a low signal-to-noise ratio, and AI-based methods are used to eliminate noise [80]. Low-resolution images can be converted into high-resolution images using deep convolutional networks, as discussed in [81]. Further, in [82], the authors discussed how the quality of MRI and CT images could be improved by using different techniques, namely “noise and artifact reduction”, “super resolution”, and “image acquisition and reconstruction”. How the two major limitations of PET imaging, namely, high noise and low-spatial resolution, are effectively handled by AI methods is discussed in [83];Image reconstruction—the process of converting raw data into images is called image reconstruction [84], and the applications of AI in the image reconstruction of MRI images have been discussed in [85];Image registration—AI methods are used in image registration or image alignment, where multiple images are aligned for spatial correspondence [86]. Further, in the case of DMRI, during image registration, along with spatial correspondence, the spatial agreement of fiber orientation among different subjects is also involved, and deep learning methods for image registration have improved accuracy and reduced computation time [87]. Improved image registration using deep learning methods for fast and accurate registration among DMRI datasets is presented in [88];Dose optimization—as discussed in [89], AI is being used in every stage of CT imaging to obtain high-quality images and help reduce noise and optimize radiation dosage [90]. Moreover, AI-based methods have found application in predicting radiation dosages, as described in [91]. AI enables the interpretation of low-dose MRI scans, which can be adopted for individuals who have kidney diseases or contrast allergies [14];Synthetic generation of CT scans—deep convolutional neural networks are useful for converting MRI images into equivalent CT images (called synthetic CT) for dose calculation and patient positioning [92]. Further, AI has been increasingly applied to problems in medical imaging, such as generating CT scans for attenuation correction, segmentation, diagnosis (of diseases), and making outcome predictions [93];Translation of EEG data—AI-based dynamic brain imaging methods that can translate EEG data in neural network circuit activity without human activity have been discussed [94];Quality assessment of MRI—a fast, automated deep neural network-based method is discussed in [95] for assessing the quality of MRIs and determining whether an image is clinically usable or if a repeated MRI is required;As described in [96], explainable AI provides reasons for the decisions in neuroimaging data.

MRI scans can have up to hundreds of layers depending upon the scan resolution, and thus manual segmentation is time-consuming, subjective, laborious, and unsuitable for large-scale neuroimaging studies [97]. Though various imaging technologies have brought mathematical and computational methods to the study of various structural, functional, diffusion-related, and other aspects, the methods have to be automated using computer-based image processing and AI-based tools. As discussed in [98], multistep planning, including the spatial realignment of individual fMRI scans, the coregistration between functional and anatomical scans, the spatial normalization of the subjects concerned, spatial smoothing, computing parametric maps, performing testing, and reporting, obviously sought the application of AI-based tools. As discussed in [99], deep learning models directly work on the raw data from EEG, which are multivariate time-series data, and extracted features with the required preprocessing and transformation, which are then used for different analytical tasks. *Brainvoyager* [100] was developed as a simple fMRI tool in the mid-1990s. It has grown as a cross-platform tool, integrating images from various devices, namely EEG, MEG, functional near-infrared spectroscopy (fNIRS), and transcranial magnetic stimulation (TMS). FreeSurfer [101] is a freely available open software package that provides automatic measurements for neuroanatomic volume, cortical thickness, surface area, and cortical gyrification (of regions of interest) throughout the brain [102]. It is widely used to obtain cortical metrics from MRI images due to its ease of configuration, accurate results, and high reproducibility [103]. In [104], the authors proposed a deep learning-based automated pipeline, *Fastsurfer,* to automatically process structural MRI. *Fast-AID Brain* is an efficient 2.5D-based deep learning tool that automatically segments a human brain MRI image into 132 cortical and noncortical regions in less than 40 s on a GPU [105]. 

### 5.5. AI in the Study of Brain Aging

CNNs help in estimating the age of a human brain from the structural MRI scan inputs [106]. The authors showed that cavities containing cerebrospinal fluid are the dominating feature in predicting the age of a brain. Brain aging is an important biomarker for identifying the risk of neurodegenerative diseases [107]. The AI models are trained to produce detailed anatomic brain maps that reveal the specific patterns of aging. Then, the biological age computed from the image is compared with the actual age. If the difference between the two is greater, then the individual is at risk of Alzheimer’s risk. 

## 6. Applications of AI for Neurological Disorders

Various neurological disorders can be broadly classified into the 7 following major categories, as discussed in [108].

Tumors;Seizure disorders;Disorders of development;Degenerative disorders;Headaches and facial pain;Cerebrovascular accidents;Neurological infections.

The applications of AI are reviewed with respect to the above categories and are presented in this section.

### 6.1. AI in Tumors

The standard method to detect a brain tumor is an MRI scan. Machine and deep learning algorithms are applied to the MRI images for three major applications, namely, tumor detection, segmentation, and grade estimation. The survey performed in [109] evidently described four types of brain tumor segmentation & classification techniques, namely, classical machine learning techniques, CNN-based techniques, capsule neural network-based techniques, and vision transformer-based techniques. Different feature extraction methods, namely first-order statistical feature extraction, gray-level co-occurrence matrix, histogram-oriented gradient, etc., have been used to extract the texture information from MRI images, and the texture information is used by multiple machine learning algorithms to classify the tumor [110]. CNN-based multigrading brain tumor classification was proposed in [111]. Different deep CNNs that classify brain tumors using user-defined hyperactive values are proposed in [112]. Multiscale grade estimation using 3D CNNs is presented in [113]. Despite the effective applications of CNN in brain tumor analysis, the accuracy of CNN is influenced by image rotations. Additionally, CNN required huge amounts of training data. These two limitations were resolved when using capsule neural network [114]-based methods. Further, due to having a smaller kernel size, CNNs are unable to extract the long-range information that is associated with image sequences that occur sequentially with respect to time, whereas the sequences are efficiently handled by vision transformer (ViT) neural networks [115].

### 6.2. AI in Seizures

Epileptic seizures develop with a sudden abnormal surge of electrical activities in the brain, and the detection of seizures is really a challenge due to the variability in their pattern. Electroencephalography recordings were analyzed using machine learning algorithms for the effective detection of seizures [116]. In [117], features from EEGs were extracted using discrete wavelet transformation and K-means clustering, and then the extracted features were analyzed using MLP for the detection of seizures. In [118], local neighbor descriptive pattern and one-dimensional local gradient pattern methods were used for extracting the features from EEG signals, and then the extracted features were analyzed using an ANN for the detection of seizures. Similarly, automatic seizure detection is discussed in [119,120]. In contrast to the above methods [119,120], which extract features from single-channel EEG data, in [121], automatic feature extraction was performed using a two-dimensional deep convolution autoencoder linked to a neural network for classifying the extracted features to detect seizures in children.

### 6.3. AI in Intellectual and Developmental Disabilities

Intellectual and developmental disabilities, such as cerebral palsy, Down syndrome, autism spectrum disorders, fragile X syndrome, attention-deficit/hyperactivity disorder, etc., begin to appear in a child’s growth, typically before the age of 18. Multimodal data, including neuroimaging data, genetic data, genomic data, electronic health records, clinical data, and behavior data (collected using different methods), form inputs for the analysis of intellectual disability (ID) and developmental disability (DD). The neuroimaging data are analyzed with a DNN to detect the presence of ID or DD in children. The presence of schizophrenia (SCZ) has been detected by a DNN by analyzing the functional connectivity pattern in fMRI data. An autoencoder-based DNN was used to analyze the fMRI data for the detection of autism [122]. In [123], an SVM algorithm was used to analyze EEG data for the detection of attention-deficit/hyperactivity disorders in children. Fragile X syndrome has an association with other medical recordings, such as circulatory, endocrine, digestive, and genitourinary factors. Based on this concept, AI-assisted screening systems have been developed to analyze the electronic health record of individuals for the detection of fragile X syndrome [124]. In [125], regarding sensitivity and specificity measures, random forest was found to outperform K-nearest neighbor, SVM, backpropagation, and deep learning in classifying Autism spectrum disorders (ASDs) in children and adolescents. The early detection of ASDs helps children at high risk undergo targeted screenings. Machine learning can detect the presence of ASDs using a toddler’s eye movements. In addition, a machine learning technique finds its role in the detection of ASDs from the presence of blood and maternal auto-antibody-based biomarkers.

### 6.4. AI in Neurodegenerative Disorders

The major neurodegenerative disorders AD, PD, and motor neuron disease pose a great challenge in that the symptoms of these diseases are not seen until a substantial number of neurons are lost [126]. Therefore, early diagnosis is difficult. MRI images can be analyzed using machine learning algorithms for the early detection of the above diseases. The authors of [127] showed that a support vector machine (SVM) could use MRI scans to successfully distinguish between individuals with AD and individuals with frontotemporal lobar degeneration (FTLD), as well as between individuals with AD and healthy individuals. In [128], 3D neural network architecture was used for the detection of AD. Patients with mild cognitive impairment (MCI) are at high risk of getting affected by AD. In [129], random forest was used to analyze MRI images for the prediction of MCI to AD conversion from one–three years before clinical diagnosis. In [130], SVM-based AD detection was performed using whole-brain anatomical MRI.

Motor neuron disease (MND), Huntington’s disease (HD), and PD are characterized by motor dysfunction. Simple tasks like drawing and handwriting can be used for the early detection of PD [131]. The authors of [132] used a combination of simple line drawings and machine learning algorithms to aid with PD diagnosis for the first time. In [133], the authors used different machine learning models, namely naïve Bayesian, decision tree, SVM, and other models, to detect the presence of AD and related dementia. As presented in [134], the individual features quantified with SVM weights as a significance map (*p-map)* were useful in enhancing the classification of dementia. Neurodegenerative diseases, such as PD, HD, and amyotrophic lateral sclerosis (ALS), produce gait disturbances, which are used as diagnostic aid for the detection of neurodegenerative diseases. Though conventional machine learning algorithms such as naïve Bayesian, SVM, and nearest neighbor algorithms help in the study of gait disturbances, they lack in considering the time information associated with gait disturbances. LSTM has been efficiently used for the study of gait disturbances [135].

### 6.5. AI in Headaches

As discussed in [136], AI is being used for detecting migraines. In [137], a four-layer XGBoostclassfier was used to analyze and classify the selfreport data of individuals into different types of headaches, including tension type headache (TTH), trigeminal autonomic cephalalgia (TAC), migraine, epicranial, and thunderclap headaches. Both machine and deep learning techniques are being used in the classification of headaches, as discussed in [138]. In the diagnosis of headaches, the symptoms and experiences of the individuals should be carefully listened to by physicians who are typically left with a busy schedule. Natural language processing and machine learning are very useful in the analysis and classification of headaches, as presented in [139].

### 6.6. AI in Cerebrovascular Accidents 

A cerebrovascular accident attack (stroke) occurs when the blood supply to the brain is reduced or interrupted. Because of a reduced or interrupted blood supply, the brain will not get oxygen, and cells of the brain become inactive. The affected people are likely to be left with permanent disability or death if the individuals are not given treatment within 3 to 4.5 h of the onset of symptoms [140]. When there is an attack, it is essential to perform an early diagnosis. There are different types of strokes, namely, ischemic stroke, hemorrhagic stroke, and subarachnoid hemorrhage, as described in [141]. An error of 20–45% is involved in the diagnosis of stroke, even for experienced doctors [142]. In [142], the authors have proposed a three-layer feed-forward ANN with a backpropagation error method for the classification of stroke. AI techniques for stroke imaging analysis were found to be encouraging [143]. The study [144] could yield 95% accuracy in the prediction of acute ischemic stroke by analyzing MRI images using a simple SVM technique. The early prediction of patients who are at high risk of acquiring malignant cerebral edema, which is the major effect following ischemic stroke, has been performed using a random forest algorithm in [145]. In [146], different machine learning algorithms, namely SVM, random forest, and ANN, were used to detect the subtype of a stroke. The assessment of stroke severity using machine learning algorithms was handled in [147].

### 6.7. AI in Neurological Infections

Though the central nervous system (CNS) is very resistant to infection by pathogens due to well-protected bony structures, once infection occurs, it is likely to progress very rapidly as the defense mechanism of the CNS is not enough to react. Despite high mortality due to various infectious diseases of the central nervous system, which are generally caused by viruses, bacteria, fungi, protozoan, etc., the detection of such diseases is very tedious due to the poor symptoms of the diseases. Computed topography and magnetic resonance imaging play a critical role in the diagnosis of neurological infectious diseases [148]. 

Viral and bacterial meningitis have similar symptoms, including a fever, headache, neck stiffness, nausea, and vomiting. The differential diagnosis between bacterial and viral meningitis is crucial, as failure to treat bacterial meningitis with proper antibiotics may lead to sequential and permanent diseases [149]. In addition, treating viral meningitis with an improper antibiotic would become an unnecessary treatment, and this may lead to changes in the microbiome and causes stress to the individuals [150]. Most of the earlier approaches for differential diagnostics use a standard method of finding an area under a curve-type analysis, in which only one predictor variable can be used to find the type of meningitis. In contrast to this kind of approach, in AI-based approaches, various predictor variables, namely cerebrospinal fluid (CSF) neutrophils, CSF lymphocytes, neutrophil-to-lymphocyte ratio (NLR), blood albumin, blood C-reactive protein (CRP), glucose, blood soluble urokinase-type plasminogen activator receptor (suPAR), and CSF lymphocytes-to-blood CRP ratio (LCR) are considered when predicting the type of meningitis, which led to higher accuracy prediction [151].

Initially, the presence of meningitis is detected from the measured values of different parameters, such as CSF and blood parameters like glucose ratio, proteins, leukocytes, etc., and then, if needed, the presence of the disease is confirmed via an invasive method like lumbar puncture. Here, the AI-based methods are useful for avoiding invasive sampling. In [152], the authors proposed an ANN-based method to detect the presence of meningitis based on temperature and various blood parameters with an accuracy of 96.67%, and thus, the proposed approach is found to serve as an alternate to invasive procedures, which involve a lengthy diagnosis time and more importantly, the individual remains unresponsive during the diagnosis, and the invasive methods produce stress in individuals. 

Neonatal sepsis with meningoencephalitis, where both meninges and the brain are infected by pathogens, has high mortality and, hence, its early detection becomes very crucial. A metabolomics approach has been practiced, whereby the metabolites that are responsible for sepsis are focused on. In addition, nuclear magnetic resonance-based metabolomics has also been used for the detection of neonatal sepsis with meningoencephalitis. Recently, AI-based techniques were used for the analysis of metabolomics data in order to achieve high accuracy. For example, in the research work [128], the authors distinguished the presence of meningoencephalitis in patients with neonatal sepsis from septic patients without meningoencephalitis using machine learning techniques.

Further, maternal infection during pregnancy is a major factor in producing long-term alterations in the functions of the fetal brain. For example, an elevated risk of autism spectrum disorders (ASDs) is found to be associated with bacterial infections [153]. Anti-N-methyl-D-aspartate receptor (anti-NMDAR) encephalitis is another common autoimmune encephalitis that exhibits symptoms including cognitive dysfunction, speech disorders, decreased consciousness, etc. Earlier studies have neglected the subtypes of the diseases. Deep learning methods can be used to develop an automated system for the prognostic prediction of the disease by extracting features from multiparametric MRI data [154].

Neuromyelitis optica spectrum disorder is an inflammatory disease of the central nervous system, and it is detected by analyzing the presence of AQP4 antibody. Traditionally, the Cox proportional-hazards model (Cox), which is a regression model, has been used for determining the association between the survival time of patients and AQP4 antibody. The Cox model has an inherent assumption that the concerned predictor variables are linearly associated with the disease, which is too simple to fit reality, where machine learning algorithms are efficient in fulfilling the need to analyze nonlinear associations [155].

Bacterial brain abscess is an intraparenchymal collection of pus [156], and it can be fatal. The common symptoms of the disease include high fever, headache, changes in consciousness, nausea, and vomiting. Cystic glioma is a kind of intracranial brain tumor whose presence is generally analyzed using CT scans and MRI images. Timely differential diagnostics should be carried out between these two diseases, as the treatments for these two are different. Along with radiomics-based features, such as the texture, size, volume, shape, and intensity characteristics of a tumor, the most complex features are extracted with the help of deep convolutional neural networks to provide physicians with more information to distinguish the above two [157].

Neurological intensive care units (ICU) are required to interpret various data, including patient demographics, clinical data, physiological waveforms, continuous electroencephalograms, laboratory tests, and images [158]. Here, machine learning techniques assist physicians in clinical decision-making by automatically interpreting heterogeneous data from different sources accurately. Neuro ICU patients are susceptible to CNS infections, which need to be detected early so that increased length of stay at Neuro ICU and increased mortality can be avoided [159]. Machine learning techniques are useful in identifying the risk factors associated with CNS infections. In [160], the authors identified four factors that are the most responsible for hospital-acquired ventriculitis and meningitis (presence of an external ventricular device, recent craniotomy, presence of superficial surgical-site infection, and CSF leaks) by using the ensemble-based XGBoost algorithm.

As discussed in [161], congenital hydrocephalus occurs in a baby born with excess fluid in the brain; this may be caused due to spina bifida or infections from its mother and is typically progressive and should be managed effectively. SVM predicts hydrocephalus by extracting the morphological features from cranial ultrasound images [162].

The usefulness of AI within the field of neuro-oncology is discussed in [163]. AI-based radiomic and radiogenomic tools give more precise initial diagnostic and therapeutic solutions for neuro-oncology. The field of radiomics extracts many quantitative features from clinical images using data-characterization algorithms, in contrast to clinical radiology, which relies on a visual assessment of images in subjective and qualitative terms [164]. Similarly, radiogenomics provides a system-level understanding of the underlying heterogeneous cellular and molecular processes. 

Neurological infectious diseases are often difficult to diagnose, and for most infections, treatments are not available for many pathogens. New infectious diseases are likely to emerge due to increased human travel all over the world, environmental changes, new pathogens, population growth, poverty, etc. Though technological advancements brought improved diagnostics and immunosuppressive therapy, neurological infectious diseases pose the following challenges:For most infections, no specific treatment is available, and the reversal of immune suppression is the only available, viable treatment;Infections can be caused by unusual pathogens, and laboratories are not equipped to detect such pathogens;Imaging techniques represent the most common diagnostic method, and the major challenge here is that most infectious diseases are likely to produce only nonspecific patterns;Many of the infected individuals may not have any symptoms, and such infections can even remain undiagnosed;Infections may be seasonal, and such infections require specialized laboratories for diagnostics;A wide range of pathogens are able to trigger immune disorders, and identifying the exact pathogen for an immune disorder is itself tedious;Prevention strategies also remain unknown in many cases;More importantly, this infectious disease can trigger other neurodegenerative and other neurological disorders.

## 7. Challenges and Future Directions

As indicated above, the convergence of AI and neuroscience has laid down a stimulating foundation for realizing intelligent applications that can predict and diagnose neurological disorders. Still, the implementation of such applications has been associated with several challenges, which lead to future research directions as well.

### 7.1. Challenges in the Creation of Interlinked Datasets Due to the Working Culture of Teams in Isolation 

Neuroscience has the potential to provide basic contributions to medicine, computing, and our understanding of human cognition. It has to adopt large-scale collaborations so that efficient resources and competencies from different teams of people can be acquired. Neuroscience has to shift from the present ‘small-scale’ working culture to large-scale teams involving experts from different domains [165]. Though neuroscience has been growing for many decades, different working teams focus on their concerned tasks in isolation. As discussed in [165], geneticists work with mice, teams working on neural microcircuitry work with rats, teams working on the visual cortex work with cats, teams working on high cognition work with monkeys and human volunteers, etc. Here, the key point is that despite the diverse teams working on different aspects, the teams are in isolation. Here the challenge is to create interlinked datasets that can reveal a better understanding of structures and cognitive functions via a holistic perspective. A more multidisciplinary approach using AI, neuroscience, and system biology is essential to create such interlinked datasets. When large teams of multiple disciplines are established, interlinked datasets can be constructed. Predictive models and simulation studies on such big data will further lead to innovations in the future.

### 7.2. Challenges Associated with Depth of Understanding in Neuro-inspired AI

Neuroscience aims to analyze a new era of large-scale high-resolution data toward identifying the loci of a disease or disorder for potential therapeutic targets, whereas neuro-inspired AI focuses on problems where human performance exceeds that of computers [166]. As the authors described, AI experts try to produce solutions at an algorithmic level rather than understanding the underlying mechanism. In this context, for AI-inspired neuroscience, identifying the right depth of understanding is a big challenge. Toward this, solutions are being implemented using a high level of abstraction (like black box solutions in DNNs). Although this looks good from an implementation point of view, from the point of interpretation or assessment, this poses a challenge.

### 7.3. Challenges Associated with the Interpretation and Assessment of AI-Based Solutions

Despite the potential ability of AI-based solutions, when such solutions are put to practical use, they have to undergo rigorous assessments. The implemented solutions are required to justify their outputs where the existing solutions are lacking [167]. In this context, the concept of explainable AI is emerging. The AI-based solutions can adopt explainable A- based interpretations for how the predictions are made. There are software tools that can be integrated with AI models. With this empowerment, neuroscience experts and doctors will gain a deeper trust in AI-based applications.

### 7.4. Challenges with Standards and Regulations

When implementing AI-based solutions for real-life applications, the first hurdle comes from the regulations. There is a lack of standards in the current regulations to assess the safety and efficacy of AI systems. In order to overcome this difficulty, the US FDA made the first attempt to provide guidance for assessing AI systems [168]. As discussed in [169], the current healthcare environment does not support data sharing. But it is mandatory for building efficient systems with a complete understanding of a wide range of aspects concerned. However, a healthcare revolution is underway to stimulate data sharing in the USA [170].

### 7.5. Methodological and Ethical Challenges

On one side, AI-based algorithms are being trained to detect the early signs of neurological disorders from various classes of data. On the other side, the practical use of such algorithms in clinical neuroscience for prediction and diagnosis is associated with various issues, as discussed in [171].

AI-based solutions are associated with inherent methodological and epistemic issues due to possible malfunctioning and uncertainty of such solutions;The over-optimization and over-fitting of AI-based solutions are likely to introduce biases in the results;There is another ethical dilemma; it is unclear whether AI models should be used to assist physicians when making decisions or should be used for automatic decision-making;Though AI-based solutions help in improving the quality of life of patients with motor and cognitive disabilities, it is also inherent that the AI-based solutions have autonomy and impact the cognitive liberty of the individuals;AI models reveal the analyzed results transparently, irrespective of the risk or sensitivity associated with the results;The training data on which the algorithms are trained will introduce a neurodiscrimination issue for the individuals concerned in the data, and this is basically due to the range of coverage of the data at hand.

### 7.6. Challenges with Neuroimaging Techniques

As discussed in [172], each neuroimaging technique has its own strengths and limitations. Different neuroimaging techniques are being combined (multimodal imaging) with the intention of obtaining complementary advantages from the individual techniques. Though multimodal imaging helps in achieving more advantages, there are still problems in its practical implementation. For example, when EEG and fMRI are combined, the magnetic resonance environment of fMRI presents some significant difficulties for safely recording meaningful EEG. Further, the magnetic fields of the MRI scanner can be disrupted by the presence of the EEG system [173,174]. In addition, another big challenge with multimodal imaging is the integration of data from various methods [175].

Further, image segmentation and common registration functions are involved when analyzing the images. Here, registration refers to aligning two or more images of the same organ acquired at different times, which may be obtained using different modalities or may be obtained from different points of view. When the images are obtained from different neuroimaging techniques, there would be variability in the appearance of the tissue or organ as the principles of image acquisition are different, and there is a lack of general rules to establish the structure correspondence [176].

Other than the technical and practical issues, patient motion, co-operation, and medical conditions also pose challenges. For example, pediatric neuroimaging is challenging due to the rapid structural, metabolic, and functional changes that occur in the developing brain. A specially trained team is needed to produce high-quality diagnostic images in children due to their small physical size and immaturity [177].

### 7.7. Challenges with Data Availability and Privacy 

Another aspect is that the performance metrics used in the algorithms do not align with clinical applications. For AI models to be used for clinical applications, they would be required to be validated with prospective data, i.e., the models should be assessed in real-world settings and with current patients in a prospective manner. In contrast, most of the models have been validated only with historical data. Unless medical data are available, the gap between the accuracy of AI and the mark of clinical effectiveness cannot be achieved. Data sharing has become very crucial. Unless data are shared among the concerned collaborating teams of experts, real innovations and values cannot be attained. In addition to data sharing, organizations related to the healthcare domain are required to follow the standards related to the healthcare domain [178]. Further, the medical organization should come forward to share the data without compromising the privacy of the concerned individuals. Fully anonymized or deidentified data can be shared according to the criteria set by the concerned standards for privacy and security [179]. For example, in the USA, the privacy and security of the healthcare data of an individual are protected by the Health Insurance Portability and Accountability Act (HIPPA) rules.

### 7.8. Challenges with Interpretation

Diagnostics errors in the interpretation of neuroimages by radiologists may happen due to the following causes [180]:Failure to consult prior studies or reports;Limitations of an imaging technique (inappropriate or incomplete protocols);Inaccurate or incomplete history;Location of lesions outside the region of interest on an image;Failure to search systematically beyond the first abnormality discovered (“satisfaction of search”);Failure to recognize a normal variant.

The combined effort of neurologists and radiologists will significantly reduce diagnostic errors [181]. 

In addition to the above challenges, the reproducibility of the results obtained by algorithms (as discussed in [182]), the biases and inequalities in algorithms, the biases from an environment, demographics, methodological flaws, and a lack of quality data should also be considered for corrective measures.

Thus, the directions for future research can be aligned toward

Making interlinked datasets from diverse and large collaborative teams from neuroscience, computing, and biology;Preparing quality data up to the standards of clinical practices;Establishing standards and regulations for data sharing;Validating AI models with prospective data;Establishing performance metrics for AI models (up to the clinical effectiveness);Developing methods and techniques for integrating data from heterogeneous neuroimaging methods;Developing software to facilitate data fusion from multimodal neuroimaging;Establishing huge data repositories for the effective training of AI algorithms (as training the algorithms with huge training data enables the model to understand the problem at hand efficiently and enhances the accuracy of the models);Bringing in interoperability standards from the different organizations involved in the neurological disease sector.

## 8. Conclusions

In this work, the complementary relationship between AI and neuroscience has been reviewed by describing how the two fields help each other. The biological neural network has brought in a shift from conventional machine learning models to deep neural network architecture, which made many real-world applications possible. The variant architectures of DNNs, along with reinforcement learning, have helped to solve complex applications, such as robot-based surgery. Additionally, neuroscience has inspired the design and development of energy-efficient spiking neural networks. In the neuroimaging field, AI has introduced tremendous changes by providing both upstream tasks, such as enhanced image acquisition, the elimination of noise, image reconstructions, image registrations, etc., and downstream tasks, such as the detection of abnormalities, characterization, diagnosis, and treatment planning. Neuroscience obtained a major benefit from AI in the analysis of complex neuroscience data. Large-scale AI-based simulations help neuroscientists to test their hypotheses, as well as to arrive at new ideas. The power of AI-based models is highly visible regarding brain data, which is phenomenal in size, speed, scope, and structure. The applications of AI in the prediction and diagnosis of various neurological disorders were reviewed, with the challenges and future directions of this area of research also highlighted. A collaborative working culture with a multidisciplinary approach will certainly make AI models efficient up to the clinical assessment level.

## Figures and Tables

**Figure 1 sensors-23-03062-f001:**
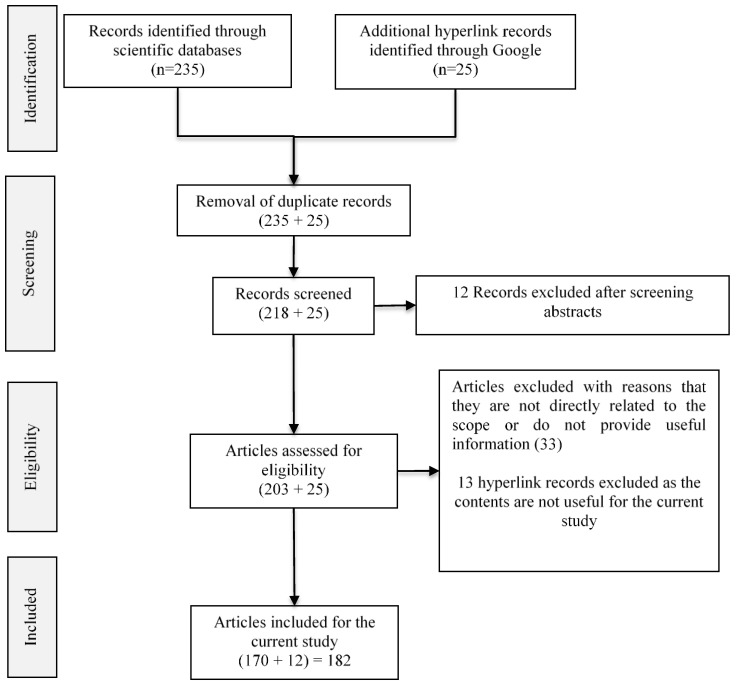
PRISMA flow diagram of the proposed review.

**Figure 2 sensors-23-03062-f002:**
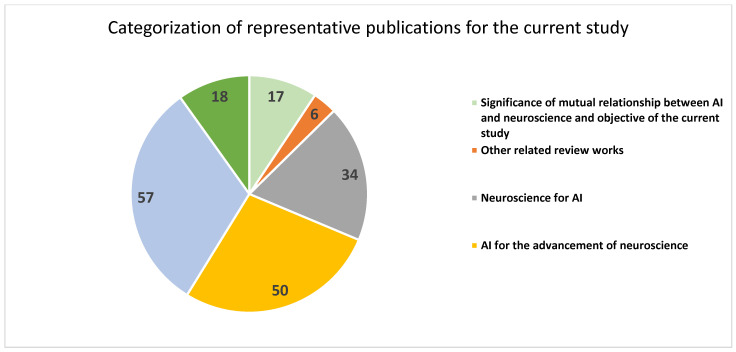
Categorization of representative publications in the current study.

**Figure 3 sensors-23-03062-f003:**
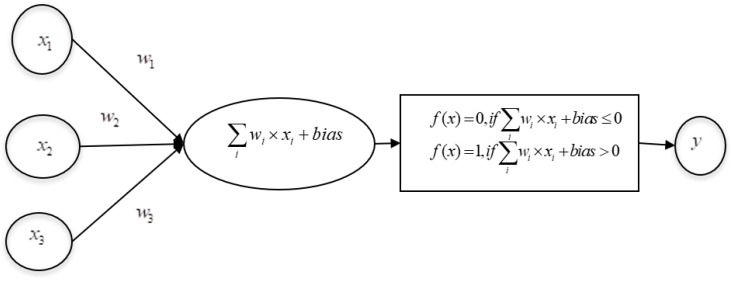
Neuro inspired perceptron classifier.

**Figure 4 sensors-23-03062-f004:**
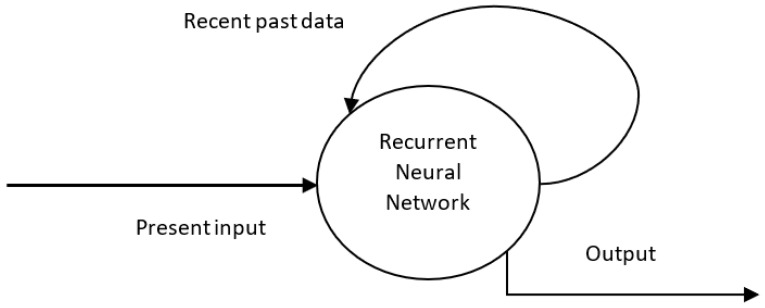
Recurrent neural network with memory of past output for the next prediction.

**Figure 5 sensors-23-03062-f005:**
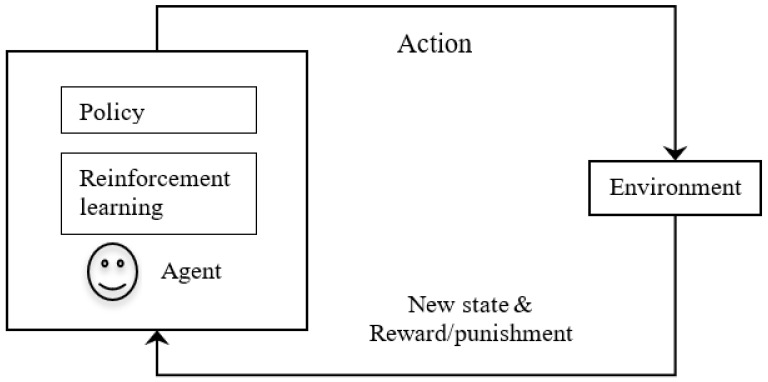
Trial-and-error-based reinforcement learning in AI.

**Figure 6 sensors-23-03062-f006:**
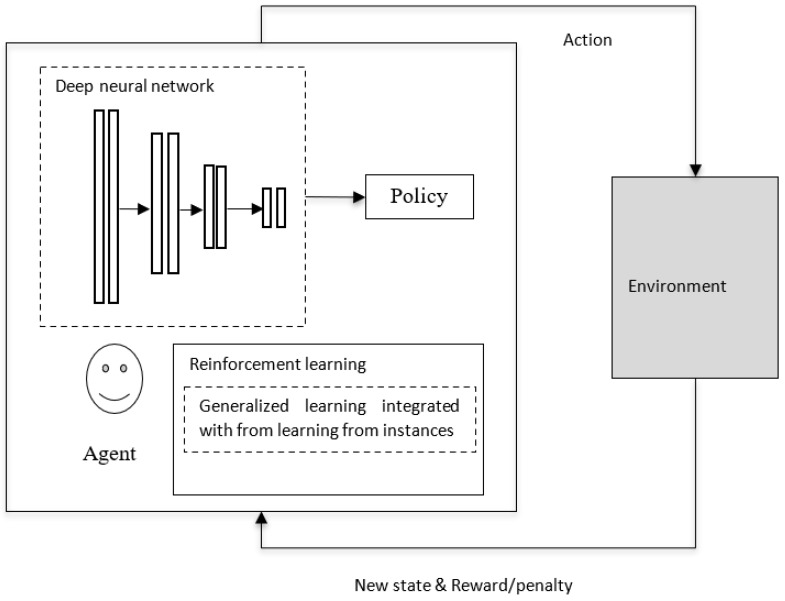
Deep reinforcement learning in AI: DNN-driven policy and neuro-inspired generalized learning integrated into learning from instances.

**Figure 7 sensors-23-03062-f007:**
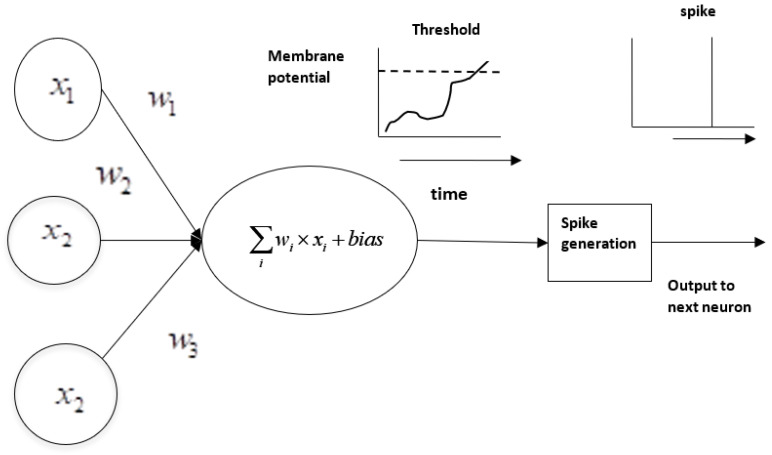
Energy-efficient spiking neural network in AI.

**Figure 8 sensors-23-03062-f008:**
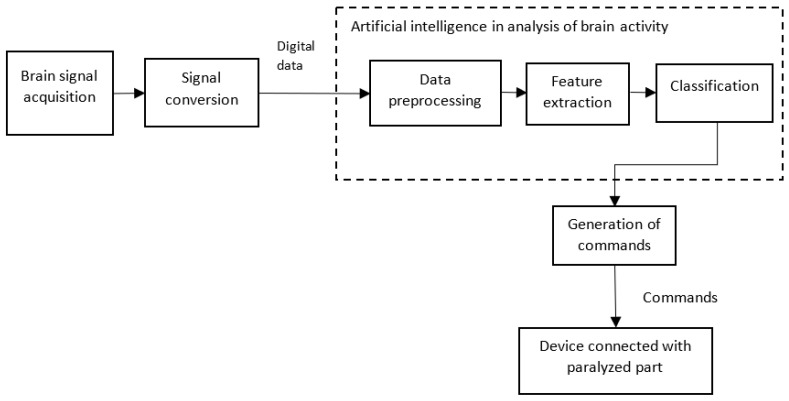
AI for the analysis of neural activity captured through a BCI for the movement of a paralyzed part.

**Table 1 sensors-23-03062-t001:** Categorization of the representative publications in the current study.

References	Specific Focus	Number of Publications
[1,2,3,4,5,6,7,8,9,10,11,12,13,14,15,16,23]	Significance of mutual relationship between AI and neuroscience and objective of the current study	17
[17,18,19,20,21,22]	Other related review works	6
[24,25,26,27,28,29,30,31,32,33,34,35,36,37,38,39,40,41,42,43,44,45,46,47,48,49,50,51,52,53,54,55,56,57]	Neuroscience for AI	34
[58,59,60,61,62,63,64,65,66,67,68,69,70,71,72,73,74,75,76,77,78,79,80,81,82,83,84,85,86,87,88,89,90,91,92,93,94,95,96,97,98,99,100,101,102,103,104,105,106,107]	AI for the advancement of neuroscience	50
[108,109,110,111,112,113,114,115,116,117,118,119,120,121,122,123,124,125,126,127,128,129,130,131,132,133,134,135,136,137,138,139,140,141,142,143,144,145,146,147,148,149,150,151,152,153,154,155,156,157,158,159,160,161,162,163,164]	AI for neurological disorders	57
[165,166,167,168,169,170,171,172,173,174,175,176,177,178,179,180,181,182]	Challenges and future directions of research	18
	Total	182

## Data Availability

Not applicable.

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
