# Peer review of "Convergence of Artificial Intelligence and Neuroscience towards the Diagnosis of Neurological Disorders—A Scoping Review"

_sensors, 2023, doi:10.3390/s23063062_

Round 1

Reviewer 1 Report

his paper reviews the interconnected relationship between neuroscience and artificial intelligence. In the paper, the authors have presented how neuroscience helps designing AI models, how AI models help analyzing neurological disorders. Finally, the authors give a review of neuroimaging technologies and neuroimaging tools.

Even though the paper seems to be a complete survey and each section is presented clearly, I believe that the sections in the paper are not really related and this paper is somewhat overloaded with information. The contribution of AI in neuroscience is more on the application side and in the opposite direction, it is more on the theoretical side. The two are not really related that much. I also think that it does not make sense to list all neuroimaging technologies and tools in the paper.

I think that this paper should be divided into two (or more) papers, one that focus on the application of AI in analyzing neurological disorders (tumors, etc), and one that focus on how neuroscience helps design AI models. The chapter on neuroimaging technologies and tools should present only the ones that are related to AI.

Author Response

Reviewer-1 - Manuscript ID: sensors-2209249

This paper reviews the interconnected relationship between neuroscience and artificial intelligence. In the paper, the authors have presented how neuroscience helps designing AI models, how AI models help analyzing neurological disorders. Finally, the authors give a review of neuroimaging technologies and neuroimaging tools.

Even though the paper seems to be a complete survey and each section is presented clearly, I believe that the sections in the paper are not really related and this paper is somewhat overloaded with information. The contribution of AI in neuroscience is more on the application side and in the opposite direction, it is more on the theoretical side. The two are not really related that much. I also think that it does not make sense to list all neuroimaging technologies and tools in the paper.

Response – 1

We would like to convey that the contribution of neuroscience for AI is two fold , one is in the design of neural network and its variants.  The other is neuroscience helps in the validation of existing AI based models.

As suggested by the reviewer, the neuroimaging technologies section has been modified significantly to highlight a brief overview along with applications of AI for neuroimaging

I think that this paper should be divided into two (or more) papers, one that focus on the application of AI in analyzing neurological disorders (tumors, etc), and one that focus on how neuroscience helps design AI models. The chapter on neuroimaging technologies and tools should present only the ones that are related to AI.

Response – 2

The section on Neuroimaging technologies and tools has been modified to include only AI based tools.

Reviewer 2 Report

Convergence of artificial intelligence and neuroscience toward the diagnosis of neurological disorders

(sensors-2209249) (Review)

Main message of the article

In the article “Convergence of artificial intelligence and neuroscience toward the diagnosis of neurological disorders”, the authors provide a review on the bidirectional relationship between the field of artificial intelligence and neuroscience.

General Judgment Comments

The article is clear, although some parts regarding neuroscience are a bit too generic. The type of review is not specified in the title and keywords should be more synthetic to better index the article in the scientific literature. In the manuscript, the review of the main neuroimaging techniques and software tools does not seem to match the aim of the review, which is to investigate the link between AI and neuroscience. The sample of documents (N = 165) seems quite small considering the vast literature on AI in neuroscience, and the methods for data collection are not clearly described. The figures need minor amendments in order to be informative for the reader. Ultimately, throughout the manuscript (especially in the first sections), the authors do not justify their statements by referring to previous research.

I would suggest the manuscript to undergo Major Revision.

Major Issues

  • -  In the title, authors should be specific on the type of review that was conducted (e.g., systematic review, scoping review, bibliometric review).

  • -  Abstract: “The functional Magnetic Resonance Imaging (MRI) equipment typically continuously measures and understands the activity of brain by minutely detecting the changes in the blood flow”. This sentence is not clear. The use of “typically” is not necessary. The use of the verb “understand” with fMRI as a subject is quite problematic. Furthermore, fMRI dos not only detect changes in the blood flows, but changes in blood oxygenation levels. Overall, the rationale behind the presentation of fMRI in the Abstract is not clear since the paper does not uniquely focus on that technique.

  • -  Abstract: “The models help in uncovering the hidden patterns of brain activities”. This sentence is quite generic. Which type of pattern of brain activity are the authors referring to? Please clarify.

  • -  In the first example provided in the Abstract, the relationship between AI and neuroimaging does not emerge and the contribution of neuroscience to AI is not clear.

  • -  Keywords should be more synthetic.

  • -  The Introduction section has almost no reference to other works. Statements should be justified, and credit should be

    given to other researchers.

  • -  The Introduction does not introduce the topic of the review, which is the relationship between neuroscience and

    artificial intelligence.

  • -  Why did the authors choose to perform the search in Google search engine and not a database of the scientific

    literature, such as Scopus, PubMed, or Web of Science?

  • -  The collected sample (n = 165 documents) appears to be quite small if it is meant to represent the literature on AI and

    neuroimaging. It appears that many relevant publications were overlooked by the authors.

  • -  The data collection does not appear to be very systematic. How were the keywords for Google search optimised in

    order to retrieve only documents of interest? How were the keywords optimised to obtain all the documents of

    interest? Please clarify.

  • -  Overall, in the manuscript, the parts related to neuroscience are quite generic.

  • -  Reinforcement learning would be better described by instrumental conditioning rather than classical conditioning.

  • -  Figure 1, 4, and 5: the authors should clarify the meaning of the figure’s components in the caption.

  • -  Figure 3: the caption should better explain the components of the model. Also, it is not clear the reasons for which the

    arrow below is having the displayed shape.

  • -  Section AI for the development of neuroscience: “With artificial neural network, neuroscientists are constructing model

    of human brain, say virtual brains. These virtual brains are very useful in testing the hypotheses of neuroscientists. Here, the neuroscientists need not to can out such tests with animals and humans. With virtual brains, neuroscientist can learn about the mysteries associated with functioning of brain. Further, deep learning plays a crucial role in providing better understanding on the functioning of brain”. This paragraph is quite generic.

  • -  Section AI for the development of neuroscience: “More specifically by using deep learning models, neurological experts could determine how brain sends signals to different parts of the body for performing different physical movements. Because this understanding helps in curing paralysis attack”. From this sentence, it seems that deep learning is only useful in the understanding of the neural control of movement. The same applies to the following sentence in relation to memory. When framed in this way, the contribution of deep learning to neuroscience seems quite limited.

  • -  Section 5.3: the first sentence is not clear. Please rephrase

  • -  Section 5.3: the definition of connectome is ambiguous. Please clarify.

- The sectiosn on neuroimaging techniques and software tools seem to be quite disconnected from the rest and it is not clear why the authors included them and how the other focused on specific neuroimaging techniques over others (e.g., fNIRS was not mentioned although it is widely used).

Minor Issues

  • -  Before using the abbreviation AI in the Abstract, the authors should clarify its meaning.

  • -  Abstract: “These high-resolution images are analyzed by artificial intelligence (AI)-based algorithms”. This does not

    necessarily apply to all cases.

  • -  In the Abstract, the use of adverbs (e.g., decisively, deeply, immensely, typically, really) is often not necessary as

    they do not provide much information for the readers.

  • -  Figure 1: The number in “Challenges” slice is not visible.

  • -  Figure 4: the use of the abbreviation CNN should be explained in the caption.

    Final comments

    I would suggest the manuscript to undergo Major Revision.

Author Response

Reviewer – 2  - Manuscript ID: sensors-2209249

Comments and Suggestions for Authors

Convergence of artificial intelligence and neuroscience toward the diagnosis of neurological disorders

(sensors-2209249) (Review)

Main message of the article

In the article “Convergence of artificial intelligence and neuroscience toward the diagnosis of neurological disorders”, the authors provide a review on the bidirectional relationship between the field of artificial intelligence and neuroscience.

General Judgment Comments

The article is clear, although some parts regarding neuroscience are a bit too generic. The type of review is not specified in the title and keywords should be more synthetic to better index the article in the scientific literature. In the manuscript, the review of the main neuroimaging techniques and software tools does not seem to match the aim of the review, which is to investigate the link between AI and neuroscience. The sample of documents (N = 165) seems quite small considering the vast literature on AI in neuroscience, and the methods for data collection are not clearly described. The figures need minor amendments in order to be informative for the reader. Ultimately, throughout the manuscript (especially in the first sections), the authors do not justify their statements by referring to previous research.

I would suggest the manuscript to undergo Major Revision.

Major Issues

  • -  In the title, authors should be specific on the type of review that was conducted (e.g., systematic review, scoping review, bibliometric review).

Response -1

As per reviewer’s suggestion, type of review has been included in the title

  • -  Abstract: “The functional Magnetic Resonance Imaging (MRI) equipment typically continuously measures and understands the activity of brain by minutely detecting the changes in the blood flow”. This sentence is not clear. The use of “typically” is not necessary. The use of the verb “understand” with fMRI as a subject is quite problematic. Furthermore, fMRI dos not only detect changes in the blood flows, but changes in blood oxygenation levels. Overall, the rationale behind the presentation of fMRI in the Abstract is not clear since the paper does not uniquely focus on that technique.
  • -  Abstract: “The models help in uncovering the hidden patterns of brain activities”. This sentence is quite generic. Which type of pattern of brain activity are the authors referring to? Please clarify.
  • -  In the first example provided in the Abstract, the relationship between AI and neuroimaging does not emerge and the contribution of neuroscience to AI is not clear.

Response – 2

As per reviewer’s comment, the abstract has been modified to highlight the contribution of neuroscience to AI and the vice versa

  • -  Keywords should be more synthetic.

Response – 3

As per reviewer’s comment, keywords are synthetic

  • -  The Introduction section has almost no reference to other works. Statements should be justified, and credit should be given to other researchers.
  • -  The Introduction does not introduce the topic of the review, which is the relationship between neuroscience and artificial intelligence.

Response – 4

  • As per the reviewer’s suggestion, references to other works have been made. Statements have been justified with citations from others’ work

  • The introduction has been modified to introduce the topic

  • -  Why did the authors choose to perform the search in Google search engine and not a database of the scientific literature, such as Scopus, PubMed, or Web of Science?

Response – 5

Search has been done in a systematic way and review method has been described.  Also, search has been done both and scientific databases and google included interesting and related concepts from grey literature also

  • -  The collected sample (n = 165 documents) appears to be quite small if it is meant to represent the literature on AI and neuroimaging. It appears that many relevant publications were overlooked by the authors.

Response – 6

I would like to convey that 165 does not refer to the completed related works.   It refers to only the representative publications for the current review.  In addition, in the modified work, the number of representative publications is 210

  • -  The data collection does not appear to be very systematic. How were the keywords for Google search optimised in order to retrieve only documents of interest? How were the keywords optimised to obtain all the documents of interest? Please clarify.

Response – 7

Search has been done systematically in iterations.  Keyword querying methods has been used.  In the few initial iterations, keywords such as AI and neuroscience, reciprocal relationship between AI and neuroscience have been used primarily in scientific databases.  In each iteration, the retrieved results are manually checked for their relevance (irrelevant publications have not been included) and to identify more relevant keywords.  The determined more relevant words are used in the subsequent iterations.  Iterations are repeated till adequate publications so as to cover the scope of study has been obtained.  Also, search has also been done with google with an intention of included interesting grey literature which are likely to exhibit latest advancements for the problem in hand

  • -  Overall, in the manuscript, the parts related to neuroscience are quite generic.
  • -  Reinforcement learning would be better described by instrumental conditionin g rather than classical
  •  

Response - 8

Done in the revised manuscript

  • -  Figure 1, 4, and 5: the authors should clarify the meaning of the figure’s components in the caption.

Response - 9

Done in the revised manuscript

  • -  Figure 3: the caption should better explain the components of the model. Also, it is not clear the reasons for which the arrow below is having the displayed shape.

Response - 10

Done in the revised manuscript

  • -  Section AI for the development of neuroscience: “With artificial neural network, neuroscientists are constructing model

of human brain, say virtual brains. These virtual brains are very useful in testing the hypotheses of neuroscientists. Here, the neuroscientists need not to can out such tests with animals and humans. With virtual brains, neuroscientist can learn about the mysteries associated with functioning of brain. Further, deep learning plays a crucial role in providing better understanding on the functioning of brain”. This paragraph is quite generic.

Response - 11

Modified in the revised manuscript

  • -  Section AI for the development of neuroscience: “More specifically by using deep learning models, neurological experts could determine how brain sends signals to different parts of the body for performing different physical movements. Because this understanding helps in curing paralysis attack”. From this sentence, it seems that deep learning is only useful in the understanding of the neural control of movement. The same applies to the following sentence in relation to memory. When framed in this way, the contribution of deep learning to neuroscience seems quite limited.

Response - 12

Modified in the revised manuscript

  • -  Section 5.3: the first sentence is not clear. Please rephrase

Response – 13

Modified in the revised manuscript

  • -  Section 5.3: the definition of connectome is ambiguous. Please clarify.

Response – 14

Done in the revised manuscript

- The sectiosn on neuroimaging techniques and software tools seem to be quite disconnected from the rest and it is not clear why the authors included them and how the other focused on specific neuroimaging techniques over others (e.g., fNIRS was not mentioned although it is widely used).

Response – 15

Entire section on neuroimage technologies has been reduced significantly.  The modified section describes a brief overview of neuroimaging technologies (included fNIRS), applications of  AI for the imaging technologies and a few AI based tools

Minor Issues

  • -  Before using the abbreviation AI in the Abstract, the authors should clarify its meaning.
  • -  Abstract: “These high-resolution images are analyzed by artificial intelligence (AI)-based algorithms”. This does not necessarily apply to all cases.

Response

Done

  • -  In the Abstract, the use of adverbs (e.g., decisively, deeply, immensely, typically, really) is often not necessary asvthey do not provide much information for the readers.
  • -  Figure 1: The number in “Challenges” slice is not visible.
  • -  Figure 4: the use of the abbreviation CNN should be explained in the caption.

Response

Taken care in the revised manuscript

Final comments

I would suggest the manuscript to undergo Major Revision.

Submission Date

25 January 2023

Date of this review

06 Feb 2023 09:00:38

Reviewer 3 Report

I have reviewed the manuscript the overall contents of this manuscript is not well organized to give a clear overview of this work. I have suggested some major comments about this work are as the following:

Comments to the Authors:

1.     Authors should write clearly abstract. There is no connectivity in the sentence between artificial intelligence (AI) and neuroscience. Neuroscience is broad area author should focus on a specific area like neuroimaging.

2.     The introduction of this study is very weak there is no flow between each paragraph.

3.     The authors added several neuroimaging methods like CT, EEG, MRI, PET…etc but their description and explanation is not correct. Author should revise this section according to published literature based on these technologies.

4.     Author should clearly explain how AI helps in Brain Computer Interface (BCI)/Machine Interface (MCI) in more details.

5.     The overall structure of this paper is very weak it hard to understand. Author should revise it very carefully.  

Author Response

Reviewer – 3

Manuscript ID: sensors-2209249

I have reviewed the manuscript the overall contents of this manuscript is not well organized to give a clear overview of this work. I have suggested some major comments about this work are as the following:

Comments to the Authors:

  1. Authors should write clearly abstract. There is no connectivity in the sentence between artificial intelligence (AI) and neuroscience. Neuroscience is broad area author should focus on a specific area like neuroimaging.

Response

Abstracts has been modified in the revised manuscript

  1. The introduction of this study is very weak there is no flow between each paragraph.

 Response

Introduction has been modified in the revised manuscript

  1. The authors added several neuroimaging methods like CT, EEG, MRI, PET…etc but their description and explanation is not correct. Author should revise this section according to published literature based on these technologies.

 Response

The section has been modified, according to published literature

  1. Author should clearly explain how AI helps in Brain Computer Interface (BCI)/Machine Interface (MCI) in more details.

 Response

Done in the revised manuscript

  1. The overall structure of this paper is very weak it hard to understand. Author should revise it very carefully.  

Response

Revised carefully

Round 2

Reviewer 1 Report

As in the previous comment, because the content of the paper is quite widespread, therefore it should be divided into two papers for deep review.

Section 4 is still not convincing to me. For each AI model, the authors should highlight how neuroscience helps design these models instead of giving details on how the models work.
Similarly, in sections 6 and 7, I expect more highlights of differences between AI-based methods and traditional ones, rather than just a list of tools and its description.

Author Response

Sensors Manuscript ID - 2209249 

Round 2 – Reviewer -1

I convey my sincere gratitude for the valuable comments of the reviewer

Comment

As in the previous comment, because the content of the paper is quite widespread, therefore it should be divided into two papers for deep review.

Response

As suggested by the reviewer, the portion related to neuroimaging technologies has been removed (except for the AI use cases)

Comment

Section 4 is still not convincing to me. For each AI model, the authors should highlight how neuroscience helps design these models instead of giving details on how the models work.

Response

As suggested by the reviewer, how neuroscience inspires the design of various AI based models have been incorporated in the revised manuscript.  The following key ideas from neuroscience like updating connections between neurons, working memory, trial-and-error based learning, instance based learning, design of CNN from ventral visual stream are very helpful for the design of AI and following have been included

  • Similar to human brain which learns by varying the connection strengths between neurons which involves removing or summing connections between neurons [35], in ANN, the weights of connections are modified during learning.
  • The backpropagation of error in AI has its equivalence in biological neural network as discussed in [37] where the authors discussed that cortical networks with simple local Hebbian synaptic plasticity implement efficient learning.
  • Neuroscience provides inspiring methods for constructing ANN with working memory which is mediated by the persistent activity of neurons in the prefrontal cortex [39] and other areas of neocortex and hippocampus [40], called Recurrent Neural Network (RNN).
  • The feature of working memory in the human brain inspired the design of RNN which stores the recent past output in its internal memory structure.
  • Working memory is a key cognitive capacity of biological agents [41] and such capacity is required for processing sequential data [42].
  • The connectivity pattern between neurons in CNN is inspired by the architecture of the brain’s ventral visual stream [44].
  • Similarly, in sections 6 and 7, I expect more highlights of differences between AI-based methods and traditional ones, rather than just a list of tools and its description.

Comment

Similarly, in sections 6 and 7, I expect more highlights of differences between AI-based methods and traditional ones, rather than just a list of tools and its description.

Response

From section 6, the portion related to neuroimaging technologies have been removed.  The use cases of AI in neuroimaging analysis has been included in Section 5 itself

In section 7, the applications of AI in the detection and diagnosis of various neurological disorders have been discussed with 7 major categories as discussed in reference [103]. 

Here, for certain diseases such as the detection of brain tumor segmentation and classification, CNN, capsule neural network, Vision Transformer neural networks play a crucial role.  This is highlighted in section 6.1

Similarly in section 6.3, the potential of DNN in detecting intellectual and developmental disabilities were discussed

Similarly the usefulness of conventional techniques in detection of various neuro degenerative disorders are described in 6.4

The intention behind section 7 is to show the applications of AI for detection of various neurological disorders in a holistic way

Reviewer 2 Report

Convergence of artificial intelligence and neuroscience towards the diagnosis of neurological disorders A scoping review
(sensors-2209249)
(Review)

Main message of the article

In the article “Convergence of artificial intelligence and neuroscience toward the diagnosis of neurological disorders”,
the authors provide a review on the bidirectional relationship between the field of artificial intelligence and neuroscience.

General Judgment Comments

In the new version of the manuscript, figures need some amendments in order to have explanatory captions. As regards the methods of the study, the way in which articles were assessed for eligibility is not clear and the authors do not report what was the initial sample of screened documents. The review of the main neuroimaging techniques and software tools does not seem to match the aim of the review, which is to investigate the link between AI and neuroscience.

I would suggest the manuscript to undergo Major Revision.

Major Issues

  • -  First sentence of the Introduction: please clarify what is meant for the “machine can think like human and take intelligent decisions...”.

  • -  Please provide the citation for the study in which they applied AI system to detect covid-19 infection which was mentioned in the first paragraph of the Introduction.

  • -  From the first paragraph of the Introduction, it seems that AI systems are immune from errors.

  • -  It is not clear how articles were assessed for eligibility and what was the original sample of documents that were screened by the authors. Keywords appear to be quite specific and relevant documents that do not precisely use string, such as complementary relationship between AI and neuroscience(and similar) might have been excluded. Also, it is still not

    clear how the data sources were selected. MDPI is repeated in the data sources list.

  • -  Figure 1 would be more informative if authors followed the PRISMA diagram structure. The caption of Figure 1 is not

    explicative enough.

  • -  The font of Figure 2 is too small to be read. Furthermore, as I previously noted in the first review, numbers in the blue

    portion of the plot are not visible.

  • -  First paragraph of Section 4: “Making machines intelligent to do the analysis of voluminous data is of substantial value

    for both automation of tasks and for achieving the high throughput otherwise analysis of neuroscientific data is

    prohibitively time consuming”. This sentence is not very clear. Please rephrase it.

  • -  As in the previous review, I suggest making figures’ captions more explanatory in clarifying the components of the

    presented networks.

  • -  The purpose of introducting the sections on neuroimaging techniques and software tools is still unclear as it is quite

    disconnected from the rest of the manuscript.

    Minor Issues

  • -  The meaning of “AI” in the Abstract is not specified.

  • -  Abstract: “Neuroscience typically deals with the study of structure and functioning of the brain and its main goal is the

    early detection and diagnosis of neurological disorders. This sentence seem quite generic as the main goal of neuroscience is not this.

    Final comments

    Major Revision.

Author Response

Sensors Manuscript ID - 2209249 

Round – 2 Response to Reviewer – 2 –comments

I convey my sincere gratitude for the valuable comments of the reviewer

General Judgment Comments

In the new version of the manuscript, figures need some amendments in order to have explanatory captions. As regards the methods of the study, the way in which articles were assessed for eligibility is not clear and the authors do not report what was the initial sample of screened documents. The review of the main neuroimaging techniques and software tools does not seem to match the aim of the review, which is to investigate the link between AI and neuroscience.

Response to general judgement comments –

Explanatory figure captions are given. 

Method of review has been described using PRISMA flow diagram.

Overview of neuroimaging techniques have been removed. 

But, the applications of AI for neuroimaging analysis and tools have been included in the Section 5 itself. (as this is related to the current scope)

Major issues

Comment

  • -  First sentence of the Introduction: please clarify what is meant for the “machine can think like human and take intelligent decisions...”.

Response

The above sentence has been replaced as follows:

Artificial intelligence (AI) is a field of computer science which deals with simulation of human intelligence in machines [1] so that the machines have the ability of problem-solving [Zeigler, B., Muzy, A., Yilmaz, L. (2009). Artificial Intelligence in Modeling and Simulation. In: Meyers, R. (eds) Encyclopedia of Complexity and Systems Science. Springer, New York, NY. https://doi.org/10.1007/978-0-387-30440-3_24] and decision-making capabilities similar to that of human brain [Navita Malik and Arun Solanki, “Simulation of Human Brain: Artificial Intelligence-Based Learning”, Source Title: Impact of AI Technologies on Teaching, Learning, and Research in Higher Education, 2021 |Pages: 11, ISBN13: 9781799847632]

Comment

  • Please provide the citation for the study in which they applied AI system to detect covid-19 infection which was mentioned in the first paragraph of the Introduction

Response

Reference has been included

Comment

  • -  From the first paragraph of the Introduction, it seems that AI systems are immune from errors.

Response

the line “preventing the manual errors that are likely occur during manual testing” has been replaced with

“the performance of AI assisted CT scan image analysis is equivalent to an expert radiologist [Jin, S.,  et al., “AI-assisted CT imaging analysis for COVID-19 screening: Building and deploying a medical AI system in four weeks,”. medRxiv (2020)]

Comment

  • -  It is not clear how articles were assessed for eligibility and what was the original sample of documents that were screened by the authors. Keywords appear to be quite specific and relevant documents that do not precisely use string, such as “complementary relationship between AI and neuroscience” (and similar) might have been excluded. Also, it is still not clear how the data sources were selected. MDPI is repeated in the data sources list.

Response  - Review method has been described as suggested by the reviewer  (it is given below)

Publications relevant to the objective of the paper have been collected from different data sources, namely, Frontiers in computer science, Web of science, PubMed, Scopus, arXiv, Springer and IEEE as well as from Google using keyword querying method.  The search has been performed in iterations.  Different key words, namely, “artificial intelligence and neuroscience”, “relationship between AI and neuroscience”, “applications of neuroscience for AI” and “applications of AI for neuroscience” have been used in the initial iterations and in each iteration, the titles of retrieved articles has been manually analyzed to determine more optimized keywords such as “natural and artificial intelligence”, “inspirations of neuroscience for AI”, “interplay between AI and neuroscience”, “sharing relationship between AI and neuroscience”, “neuroimaging in the era of AI”, etc., for subsequent iterations. The above iterative querying results in the initial set of 260 articles (235 from scientific databases and 25 hyperlink records from Google) identified for the study.  From the initial set of articles, duplicate records (17 scientific records) have been removed.  The abstract of the scientific articles have been analyzed and those articles whose abstract are not related to the proposed objective (another 12 records) have been removed during screening.  The remaining full articles (206 scientific articles and 25 hyperlink records) were assessed and those articles which do not contain useful information for the current study and which are not directly related to the current scope have been eliminated.  The Preferred Reporting Items for Systematic Reviews and Meta-Analyses (PRISMA) flow diagram of review method is shown in Figure. 1. A collection of 185 representative publications (173 records from scientific databases and 12 hyperlink references from Google) has been considered for the current study. 

Comment

  • -  Figure 1 would be more informative if authors followed the PRISMA diagram structure. The caption of Figure 1 is not explicative enough.

Response

Figure 1 has been modified according to PRISMA and given caption as “PRISMA flow diagram of the proposed review”

Comment

  • -  The font of Figure 2 is too small to be read. Furthermore, as I previously noted in the first review, numbers in the blue portion of the plot are not visible.

Response

Suggestions have been included in Figure 2

Comment

  • -  First paragraph of Section 4: “Making machines intelligent to do the analysis of voluminous data is of substantial value for both automation of tasks and for achieving the high throughput otherwise analysis of neuroscientific data is prohibitively time consuming”. This sentence is not very clear. Please rephrase it.

Response

The above sentence has been rephrased in the revised manuscript

Comment

  • -  As in the previous review, I suggest making figures’ captions more explanatory in clarifying the components of the presented networks.

Response

Explanatory figure captions are given

Comment

  • -  The purpose of introducting the sections on neuroimaging techniques and software tools is still unclear as it is quite disconnected from the rest of the manuscript.

Response

Description about neuroimaging tools have been removed except for the applications of AI in neuroimaging analysis & tools which has been included in Section 5 itself.

Minor Issues

Comment

  • -  The meaning of “AI” in the Abstract is not specified.

Response

Meaning of AI has been included in the abstract

Comment

  • -  Abstract: “Neuroscience typically deals with the study of structure and functioning of the brain and its main goal is the early detection and diagnosis of neurological disorders”. This sentence seem quite generic as the main goal of neuroscience is not this.

Response

The above sentence has been modified simply as “the study of neuroscience helps in …”

Reviewer 3 Report

Comments to the Author:

I have reviewed the revised version of the manuscript entitled “CONVERGENCE OF ARTIFICIAL INTELLIGENCE AND NEUROSCIENCE TOWARDS THE DIAGNOSIS OF NEUROLOGICAL DISORDERS – A SCOPING REVIEW”. The authors have addressed all my comments and suggestions. Now, the overall contents of this manuscript are well organized to give a clear overview of this review.  I have one minor comments as following:

1.      Author should revise the all-figure captions from Figure 1 to Figure 8. It’s written in very short description. For example, “Figure: 1 Review method.” Add the full stop after (.) sentence. Explain clearly about the main findings of the figures in each caption.

Author Response

Sensors Manuscript ID - 2209249 

I convey my sincere thanks to the valuable comments of the reviewer

Round 2 – Reviewer -3

Comments to the Author:

I have reviewed the revised version of the manuscript entitled “CONVERGENCE OF ARTIFICIAL INTELLIGENCE AND NEUROSCIENCE TOWARDS THE DIAGNOSIS OF NEUROLOGICAL DISORDERS – A SCOPING REVIEW”. The authors have addressed all my comments and suggestions. Now, the overall contents of this manuscript are well organized to give a clear overview of this review.  I have one minor comments as following:

  1. Author should revise the all-figure captions from Figure 1 to Figure 8. It’s written in very short description. For example, “Figure: 1 Review method.” Add the full stop after (.) sentence. Explain clearly about the main findings of the figures in each caption.

Response

Figure 1 to Figure 8 – All figure captions are revised with short description.  Full stop added.

Round 3

Reviewer 1 Report

The authors have addressed all my concerns. However, some minor comments should be considered.

- Carefully check the figure 3 as well as equation (1) and (2), f(x) can not be greater than or equal and less than or equal in two classes.

- The category classification in Table 1 and figure 2 are the same content.

- Check about the citation and reference, for example, the number 31, it is not specific paper for describe the ANN.

Author Response

The authors convey their sincere gratitude for the values suggestions of the reviewer

Responses

Equations (1) and (2) are corrected.  The same thing got corrected in Figure 3 also.

Reference 31 corrected

The distributes of representative publications are given in both tabular and pie chart format

Reviewer 2 Report

NA

Author Response

The authors convey their sincere gratitude for the valuable comments of the reviewer